

# Rapid point-of-care detection of BK virus in urine by an HFman probe-based loop-mediated isothermal amplification assay and a finger-driven microfluidic chip

Yongjuan Zhao[1,*], Yi Zeng[1,*], Renfei Lu[2], Zhiying Wang[3], Xiaoling Zhang[1], Nannan Wu[1], Tongyu Zhu[4,5], Yang Wang[3] and Chiyu Zhang[1]

[1] Shanghai Public Health Clinical Center, Shanghai, China
[2] Nantong Third Hospital Affiliated to Nantong University, Nantong, China
[3] Beijing Advanced Innovation Center for Biomedical Engineering, Key Laboratory for Biomechanics and Mechanobiology, School of Engineering Medicine, Beihang University, Beijing, China
[4] Shanghai Medical College, Shanghai, China
[5] Zhongshan Hospital, Shanghai, China
[*] These authors contributed equally to this work.

Corresponding authors
Yang Wang, w_yang89@163.com,
wangyang2022@buaa.edu.cn
Chiyu Zhang,
chiyu_zhang1999@163.com

## ABSTRACT

**Background.** BK virus (BKV)-associated nephropathy (BKVN) is one of the leading causes of renal dysfunction and graft loss in renal transplant recipients. Early monitoring of BKV in urine is crucial to minimize the deleterious effects caused by this virus on preservation of graft function.

**Methods.** We report a simple, rapid, sensitive loop-mediated isothermal amplification (LAMP) assay using an HFman probe for detecting BKV in urine. To evaluate the performance of the assay, a comparison of the HFman probe-based LAMP (HF-LAMP) assay with two qPCR assays was performed using urine samples from 132 HIV-1 infected individuals. We further evaluated the performance of HF-LAMP directly using the urine samples from these HIV-1 infected individuals and 30 kidney transplant recipients without DNA extraction. Furthermore, we combined the HF-LAMP assay with a portable finger-driven microfluidic chip for point-of-care testing (POCT).

**Results.** The assay has high specificity and sensitivity with a limit of detection (LOD) of 12 copies/reaction and can be completed within 30 min. When the DNA was extracted, the HF-LAMP assay showed an equivalent and potentially even higher sensitivity (93.5%) than the qPCR assays (74.2–87.1%) for 132 urine samples from HIV-1 infected individuals. The HF-LAMP assay can be applied in an extraction-free format and can be completed within 45 min using a simple heat block. Although some decreased performance was seen on urine samples from HIV-1 infected individuals, the sensitivity, specificity, and accuracy of the extraction-free BKV HF-LAMP assay were 95%, 100%, and 96.7% for 30 clinical urine samples from kidney transplant recipients, respectively.

**Conclusion.** The assay has high specificity and sensitivity. Combined with a portable finger-driven microfluidic chip for easy detection, this method shows great potential for POCT detection of BKV.

## INTRODUCTION

BK virus (BKV), which belongs to the Polyomaviridae family, is a small nonenveloped icosahedral virus with circular doubled-stranded DNA and a genome size of approximately 5,000 bases (*Alcendor, 2019*). BKV infects and establishes latency primarily in the urothelium during childhood. BKV is classified into four subtypes (I, II, III, and IV), and subtype I is further subdivided into four subgroups (Ia, Ib1, Ib2, and Ic) based on variation in the VP1 gene sequence (*Hussain et al., 2020*). The most common BKV type worldwide is genotype I (approximately 80%), followed by genotype IV (approximately 15%). Up to 80–90% of adults are seropositive for BKV (*Antonsson et al., 2010*; *Egli et al., 2009*; *Kamminga et al., 2018*). Usually, BKV infection is asymptomatic, but some infections are accompanied by mild respiratory illness or urinary tract disease (*Mischitelli et al., 2008*). BKV can establish a persistent infection in the urinary tract (*Furmaga et al., 2022*). Latent BKV infection can be reactivated in immunosuppressed individuals and result in related diseases, such as viremia, hemorrhagic cystitis, ureteral stricture, encephalitis, nephritis, retinitis, pneumonia, and even multiple-system organ failure (*Alcendor, 2019*; *Chong et al., 2019*). Most notably, BKV-associated nephropathy (BKVN) is one of the leading causes of renal dysfunction and graft loss in renal transplant recipients. The prevalence of BKVN in renal transplant patients is approximately 10%. Because BKV infection is often asymptomatic at an early stage and there lacks effective antiviral treatments, the risk of graft loss is up to 80% (*Mallavarapu et al., 2021*; *Teutsch et al., 2015*). Currently, the mainstay treatment of BKVN is reduction of immunosuppression (*Teutsch et al., 2015*) until the development of specific antiviral treatment strategies. Therefore, the development of a rapid and simple point-of-care testing (POCT) method for the detection of BKV is crucial and necessary to minimize the deleterious effects caused by BKV infection on the preservation of graft function (*Bateman et al., 2017*; *Gouvea et al., 2016*).

Various molecular approaches have been previously developed for the detection of BKV infection. As a gold standard of nucleic acid diagnosis, quantitative polymerase chain reaction (qPCR) is widely used to monitor BKV infection (*Dumoulin & Hirsch, 2011*; *Hussain et al., 2020*; *Kato et al., 2017*). However, qPCR has some limitations; for example, it requires precise thermal-cycling machines and highly trained personnel, and it is relatively time-consuming (often taking 1−1.5 h), which limit its application for POCT in resource-limited areas. Only approximately 50% of all transplant recipients are evaluated for BK virus infection in resource-limited regions because PCR-based screening assays are expensive and there is limited funding with no medical insurance (*Bagchi et al., 2017*; *Yooprasert & Rotjanapan, 2018*). Hence, a rapid and convenient POCT assay for BKV detection is necessary and will be helpful for decreasing BKVN and transplant complications.

POCT is a promising trend for molecular diagnostics. In particular, the current outbreak of COVID-19 has brought about realization of the necessity and importance of POCT diagnosis, and many nucleic acid POCT methods have been developed (*Augustine et al., 2020*; *Kang et al., 2022*). The loop-mediated isothermal amplification (LAMP) method is a novel nucleic acid amplification method developed in 2000 that uses four primers to

recognize six regions of the target DNA (*Notomi et al., 2000*). LAMP is capable of detecting even a few copies of a target nucleic acid under isothermal conditions (usually 60−65 °C) using strand-displacing DNA polymerase (*Nagamine, Hase & Notomi, 2002*; *Notomi et al., 2000*). Unlike PCR, LAMP can be performed in a low-resource setting using a simple heating device. A LAMP assay using non-sequence-specific fluorescent dye SYBR green was previously developed for BKV detection and showed a sensitivity of 100 copies per reaction (*Bista et al., 2007*). However, frequent nonspecific amplification largely limit its clinical application. Recently, we developed a multiplexed variant-tolerant real-time LAMP method that utilizes a small amount of additional high-fidelity DNA polymerase to mediate the release of fluorescence signals from a probe (*e.g.*, an HFman probe) (*Dong et al., 2022*; *Zeng et al., 2022*). High-fidelity DNA polymerase has a $3'-5'$ exonuclease (proofreading) activity to remove $3'$-misincorporated nucleotide of newly synthesized DNA or mismatched base in the primer. The use of the HFman probe and high-fidelity DNA polymerase largely improves the specificity of LAMP. Furthermore, this HFman-probe-based LAMP method has comparable performance to qPCR with regard to specificity and sensitivity and a faster reaction speed (approximately 30 min) than the latter (*Dong et al., 2022*).

In this study, we applied this technique to develop an HFman probe-based LAMP (HF-LAMP) assay for BKV detection. The HF-LAMP assay exhibited high specificity and sensitivity and had very good consistency (98.5%) with the qPCR assay in the detection of 132 HIV-1 infected clinical samples. Furthermore, we further validated the POCT performance of this assay directly using urine (*i.e.,* without DNA extraction) in combination with a portable finger-driven microfluidic chip.

## MATERIALS & METHODS

### Primer design

Primers and probes specific for the VP1 gene of BKV were designed using Primer Explorer V5 software (Eiken Chemical Co., Ltd., Tokyo, Japan; https://primerexplorer.jp/e/) after multiple alignment using MEGA7 (a robust Molecular Evolutionary Genetics Analysis software; https://www.megasoftware.net/). Five sets of primers were designed, and each set included two outer primers (F3 and B3), two inner primers (FIP and BIP), and loop primers (LF and/or LB). The primer sequences are shown in Table S1. All primers and probes were synthesized by Sangon Biotech (Shanghai, China).

### Preparation of BKV standard

PUC-57 plasmids containing the VP1 gene (GenBank accession no: AB298947.1, 1610–2363 nt) of BKV were synthesized by Sangon Biotech (Shanghai, China) as standards. The concentrations of the plasmid standards were quantitated with a Qubit® 4.0 Fluorometer (Thermo Fisher Scientific, Waltham, MA, USA). The copy numbers of the plasmids were calculated using the following formula: dsDNA copies/mL = (DNA concentration (g/mL)/(DNA length $\times 660$)) $\times 6.022 \times 10^{23}$.

## Reaction system of the HF-LAMP assay

The reaction of the HF-LAMP was established according to our previous paper (*Dong et al., 2022*). The 25 µL reaction mixture contains 3 µL of template, 8 U of Bst 4.0 DNA polymerase (Haigene, Haerbin, China), 0.15 U of Q5® high-fidelity DNA polymerase (NEB, Beijing, China), 2.5 µL of 10× $Mg^{2+}$-free isothermal amplification buffer (Haigene, Haerbin, China), 8 mM $MgSO_4$, 1.8 mM of each dNTP, 0.1 µM each of F3 and B3, 1.0 µM each of FIP and BIP, 0.2 µM LB and 0.2 µM HFman probe (or LF if used). The reaction was performed at 64 °C for 50 min in a CFX96 Touch Real-Time PCR detection system (Bio-Rad Laboratories, Hercules, CA, USA).

## Evaluation of the specificity and sensitivity of the LAMP assay

The specificity of the LAMP system was evaluated using nine common viruses, including human JC polyomavirus (JCV); cytomegalovirus (CMV); papillomavirus types 1, 2, 3, and 4 (HPV-1, HPV-2, HPV-3, HPV-4); hepatitis B virus type 16 (HBV-16); adenoviral type 5 (Adhu5); and human immunodeficiency virus type 1 (HIV-1). Nucleic acids of the nine viruses (50–200 ng/µL) were obtained from positive clinical samples. To evaluate the sensitivity of the LAMP system, the BKV standard was ten-fold serially diluted (from $10^4$ to $10^0$ copies/µL), and each dilution was used for the reaction.

The limit of detection (LOD) is the lowest detectable concentration at which around 95% of all true positive replicates test positive. To determine the LOD, five-fold serial dilution of the DNA standard (3,000 copies, 600 copies, 120 copies, 24 copies and five copies) were used in the 25 µL reaction. Each dilution was tested in a set of 20 replicates. The LOD was defined as a 95% probability of obtaining a positive result, using the probit regression analysis implemented in SPSS 17.0 software (SPSS, Inc., Chicago, IL, USA).

## Nucleic acid extraction

Nucleic acid was extracted according to the manufacturer's instructions of a commercial BKV qPCR kit (Sinomd Bio, Beijing, China). Fifty microlitres of lysis buffer was added to the precipitates obtained from 1 mL urine samples by centrifugation. The lysate was vortexed and incubated at 65 °C for 30 min. The supernatant was used for nucleic acid detection after centrifugation.

## Reaction with a commercial BKV qPCR kit (qPCR assay 1)

Reactions with a commercial BKV qPCR kit (Sinomd Bio, Beijing, China) were performed according to the manufacturer's instructions using a CFX96 Touch Real-Time PCR detection system (Bio-Rad Laboratories, Hercules, CA, USA). Five microlitres of template solution was used for the reaction. The thermal cycling protocol was 1 min at 95 °C for initial denaturation followed by 40 cycles of 5 s at 95 °C for denaturation and 15 s at 60 °C for annealing and extension. The threshold cycle (Ct) of the qPCR assay was determined by the detection system as the cycle at which sample fluorescence crosses the threshold.

## In house qPCR assay (qPCR assay 2)

In house qPCR was performed using Premix Ex Taq™ (Probe qPCR) Kit (Takara Bio Inc. Shiga, Japan) in a CFX96 Touch Real-Time PCR detection system (Bio-Rad Laboratories).

The primer sequences used in the qPCR assay have been described previously (*Leung et al., 2001*). The 20 µL qPCR mixture contained 10 µL of 2× Premix, 1 µL of 10 µM BKV forward primer (5′- AGTGGATGGGCAGCCTATGTA-3′), 1 µL of 10 µM BKV reverse primer (5′-TCATATCTGGGTCCCCTGGA-3′), 0.5 µL of 10 µM hydrolysis probe (5′-FAM- AGGTAGAAGAGGTTAGGGTGTTTGATGGCACAG-BHQ1-3′), and 5 µL of template solution. The thermal cycling protocol was 1 min at 95 °C for initial denaturation followed by 40 cycles of 5 s at 95 °C for denaturation and 15 s at 60 °C for annealing and extension.

## Evaluation of BKV HF-LAMP using clinical samples

The clinical sample collection was approved by Shanghai Public Health Clinical Center ethics committee (2018Y032) and registered and recorded at the Chinese Clinical Trial Registry (ChiCTR1800017947). The requirement for informed consent was waived because the clinical urine samples were the rest samples after necessary clinical tests. A total of 132 urine samples from HIV-1 infected individuals were collected at the Clinical Laboratory of the Nantong Third Hospital affiliated with Nantong University Hospital from 2021.9.1 to 2021.9.16. In addition, clinical urine samples from kidney transplant patients were obtained from Shanghai Public Health Clinical Center, including 20 BKV positive samples and 10 BKV negative samples. The status of BKV infection of the kidney transplant patients were previously determined by a digital PCR assay (*Xu et al., 2021*). The samples were divided into aliquots and stored at −80 °C to avoid frequent freeze-thaw cycles. A commercial BKV qPCR kit (Sinomd Bio, Beijing, China), a reported qPCR assay (*Leung et al., 2001*), and HF-LAMP were applied to analyse the urine specimens in parallel. The concordance rate between both assays was calculated with the following formula: (number of consistent results by both methods/total number) ×100%.

## DNA extraction-free HF-LAMP

Fifty microlitres of urine samples collected from HIV-1 infected individuals or kidney transplant patients was incubated at 95 °C for 3 min (Gandasegui et al. 2015). HF-LAMP was performed directly using 5 µL of urine. The results were compared with those from the qPCR assay 2. The concordance rate between both assays was calculated with the following formula: (number of consistent results by both methods/total number) ×100%.

## Portable finger-driven microfluidic chip

A portable finger-driven microfluidic chip (Fd-MC) was fabricated according to a previous report with slight modifications (*Wang et al., 2022*). The number of reaction chambers was changed from five to two. The volume of waste reservoir been modified for better detection of BK virus from urine samples. Moreover, we integrated the HF-LAMP system in the chip for POCT diagnosis. Integrating with HF-LAMP, 37 urine samples including 27 BKV positive samples and 10 negative samples that were confirmed by the qPCR assay 2, selected from 132 urine samples of HIV-1 infected individuals, and were subjected to the experiments after heating at 95 °C for 3 min by the portable finger-driven microfluidic chip. During manufacturing, the DNA washing buffer (350 µL) and the LAMP buffer (25 µL) were pre-packaged into the liquid storage chambers on the substrate. The DNA washing

buffer was 70% ethanol in 10 mM Tris (pH =8), primers and FAM-labeled Hfman probe targeting for the BKV, and RNase-free water (no template control, NTC) were all added to the reaction chambers. LAMP reactions would be initiated once the reaction buffer and the sample containing targets were driven to the sealed chamber by the pressure generated by the finger-actuated operation. Subsequently, the Fd-MC was heated in a water bath at 64 °C for 50 min and the results were read directly by naked eye with an orange filter (Luyor LUV-30A; Luyor, Shanghai, China) under a hand-held blue lamp (Luyor 3260; Luyor, Shanghai, China).

## RESULTS

### Primer design for the HF-LAMP assay

The VP1 coding region, which is the most conserved genomic region of the virus (GenBank: AB298947.1, 1,610–2,363 bp), shows very high identity (over 95%) in all genotypes of BKV (*Furmaga et al., 2021*). Accordingly, LAMP primers were designed to amplify the VP1 gene (Fig. 1), which is also the target region of most PCR assays (*Xu et al., 2021*). For better detection of BKV, five sets of primers were designed based on VP1 gene. To investigate the properties of different sets of primers and thus select the optimal set, all primer sets were subjected to a comparative experiment under the same conditions (reaction system and program, as well as template input). The results indicated that all five sets of primers can be used for amplification of the VP1 gene of BKV (Fig. S1). Compared to the other four sets of primers, primer set BKV-2 showed higher amplification efficiency with a relatively lower time threshold (Tt) value, which was selected for further use. The primer set BKV-2's information and sequence alignments of the primer and probe regions are shown in Fig. 1 and Fig. S2, respectively.

The HFman probe can be designed with identical sequence to LF or LB. In this study, it was synthesized according to LB sequence, and labelled with a Cy5 fluorophore and a BHQ2 quencher group at the 3′- and 5′-ends (or FAM and BHQ1 at 3′- and 5′-ends), respectively. The 3′ fluorescence-labelled base of the HFman probe can be recognized and removed by the high-fidelity DNA polymerase to release fluorescent signal, as previously demonstrated (*Dong et al., 2022*).

### Specificity and sensitivity of the BKV HF-LAMP assay

The specificity of the BKV HF-LAMP assay was assessed using nine common human viruses, including JCV, CMV, HPV-1, HPV-2, HPV-3, HPV-4, HBV-16, Adhu5, and HIV-1. Except BKV, no amplification curves were observed in any of the other nine common human viruses, indicating a high specificity of the assay (Fig. 2A). The sensitivity was determined using standards serially diluted 10-fold from $10^4$ to $10^3$, $10^2$, 10, and 1 copies/μL. Three microlitres of template was added to each LAMP reaction. The detection ability of the LAMP assay was three copies/reaction in less than 20 min (Fig. 2B). We further measured the LOD of the HF-LAMP assay. The LOD was estimated as 12 copies per 25 μL reaction and all 20 negative controls were tested at same time without amplification (Table 1).
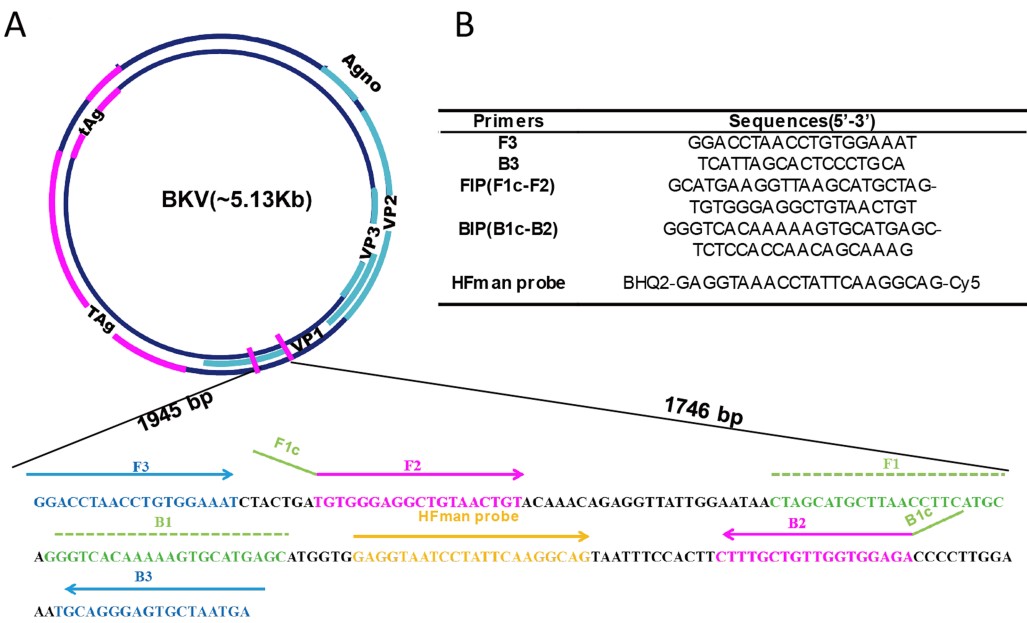

**Figure 1 Primer information.** Locations (A) and sequences (B) of the primers and probe in the BKV genome. Broad lines indicate open reading frames. Broad purple lines indicate the early region which is predominantly expressed before viral DNA replication to encodes the large T-antigen (T-ag) and small t-antigen (t-ag); broad blue lines indicate the late region which is expressed late in the viral life cycle to encodes the agnoprotein and capsid proteins VP1, VP2, and VP3.

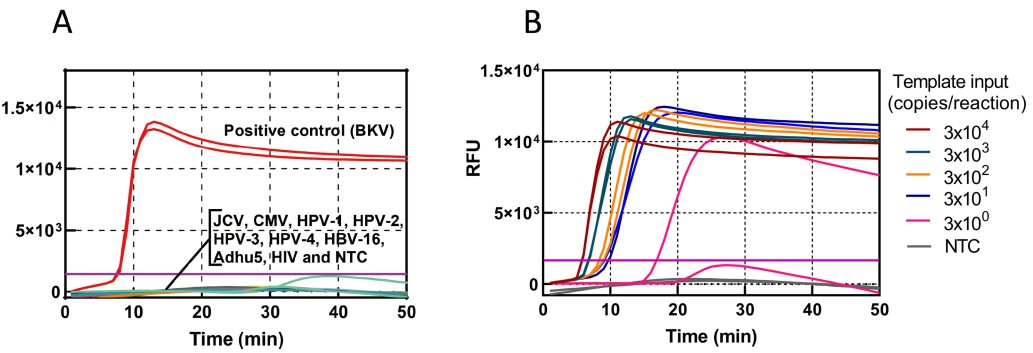

**Figure 2 Sensitivity and specificity of the HF-LAMP assay.** (A) Specificity of the BKV HF-LAMP assay. Approximately 3000 DNA copies of BKV were used as the positive control. The testing viruses included JCV, CMV, HPV-1, HPV-2, HPV-3, HPV-4, HBV-16, Adhu5, and HIV. (B) Sensitivity of the BKV HF-LAMP assay. The sensitivity of detection was determined with 10-fold serial dilutions of the BKV standard from $3 \times 10^4$ to $3 \times 10^0$ copies/reaction. NTC: no-template control, purple horizontal line: threshold.

## Evaluation of HF-LAMP using extracted DNA from clinical samples of HIV-1-infected patients

HIV-1 infected individuals are an immune-impaired population and often have a relatively higher BKV-positive rate than other populations (*Hu et al., 2018*). To verify the clinical

**Table 1  Limit of Detection (LOD) of the HF-LAMP for BKV detection.**

| Template input (copies/25 μL reaction) | 3000 | 600 | 120 | 24 | 5 | LOD (copies/25 μL reaction) |
|---|---|---|---|---|---|---|
| BKV (positive/total) | 20/20 | 20/20 | 20/20 | 20/20 | 9/20 | 11.6 |

**Table 2  Comparison of the DNA extraction HF-LAMP assay with two qPCR assays from 132 clinical samples from HIV-1 infected patients.**

| Method | | qPCR assay 1 (commercial qPCR kit, Sinomd) | | | qPCR assay 2 (previously reported method) | | | | |
|---|---|---|---|---|---|---|---|---|---|
| | Items | Pos. | Neg. | Concordance rate (%) | Pos. | Neg. | Concordance rate (%) | Total | Positive rate (%) |
| HF-LAMP assay | Pos. | 21 | 8 | 92.4 | 27 | 2 | 98.5% | 29 | 22.0 |
| | Neg. | 2 | 101 | | 0 | 103 | | 103 | |
| Total | | 23 | 109 | | 27 | 105 | | 132 | |
| Positive rate (%) | | 17.4 | | | 20.5 | | | | |

**Notes.**
Pos., positive; Neg., negative.

applicability of the BKV HF-LAMP assay, a total of 132 urine samples were collected from 132 HIV-1 infected patients. For comparison, an approved commercial qPCR kit (Sinomd Bio, Beijing, China) (qPCR assay 1) and a previously reported qPCR assay (qPCR assay 2) (*Leung et al., 2001*) were used in parallel. The BKV positive rate among this cohort was 18.9%–22.0% according to the three different assays (Table 2). The BKV positive rate was higher by the HF-LAMP assay (22.0%) than both qPCR assays (18.9% and 20.5%) (Table 2). In particular, all positives by the HF-LAMP assay could be confirmed by at least one qPCR assay. Therefore, two samples that were tested negative by the HF-LAMP assay, were tested positive by the qPCR assay 1 (Fig. 3). If the positive samples by any two of the three assays (or any one of the two qPCR assays) are considered as true positive, the HF-LAMP assay has a higher detection sensitivity (93.5%) than both qPCR assays (74.2% and 87.1%). In addition, the HF-LAMP assay showed a higher concordance rate with the qPCR assay 2 (98.5%, 130/132) than the qPCR assay 1 (Fig. 3B).

The Ct values of both qPCR assays ranged from 16.9 to 36.1 for detected positive samples, while the Tt values of the HF-LAMP assay ranged from 3.9 to 15.4 min (Fig. 3A). In particular, HF-LAMP assay generated positive signals within approximately 15 min for most samples (25/29) (Fig. 3A), while the qPCR assay usually requires at least 1 h for extracted DNA. The results indicate that the LAMP assay was substantially faster than the qPCR assays.

## Clinical evaluation of the nucleic acid extraction-free BKV HF-LAMP assay using clinical samples from HIV-1 infected patients

To further simplify the HF-LAMP assay, we developed an easy-to-operate workflow (Fig. 4A) and further evaluated the performance of HF-LAMP directly using HIV-1 infected clinical samples without extraction. The urine was incubated at 95 °C for 3 min

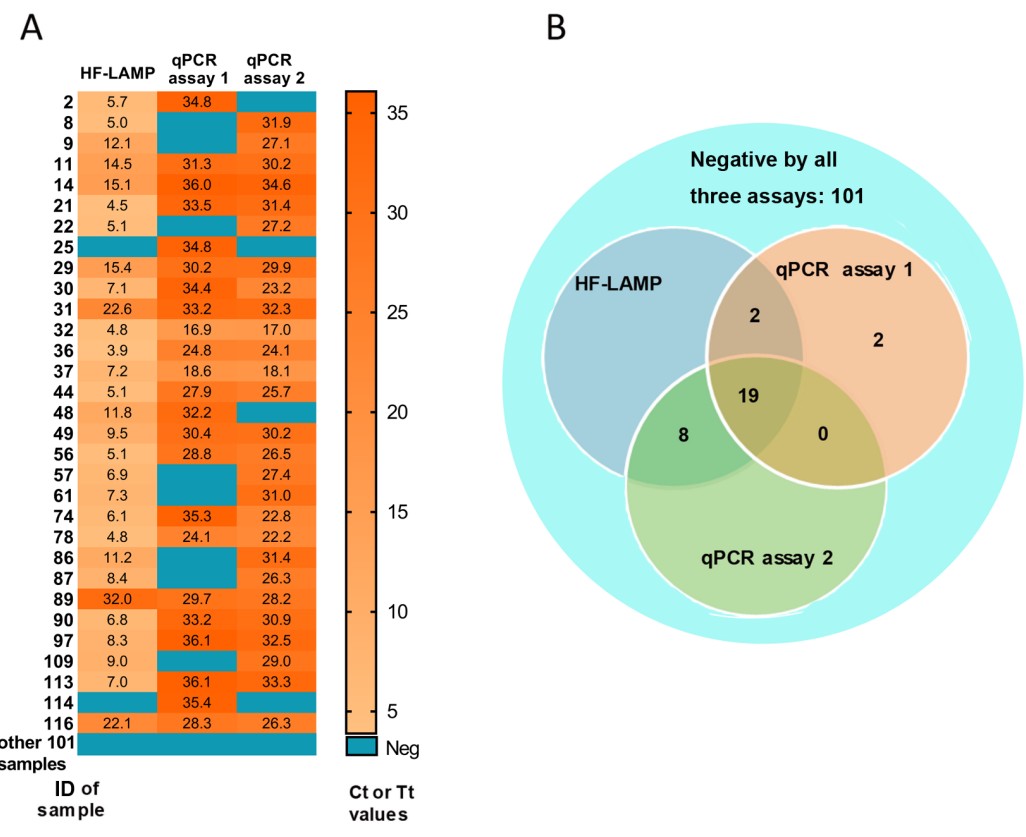

**Figure 3** **Clinical validation of BKV HF-LAMP using extracted DNA from urine.** (A) Heatmap of Tt values of the HF-LAMP assay and Ct values of two qPCR assays on 132 urine samples using extracted DNA. (B) Venn diagram of the number of samples detected by different assays based on the result of qPCR assays 1, the commercial qPCR kit; qPCR assays 2, the previously reported qPCR assay.

to lyse the viral particles and release nucleic acids. Then, the urine was directly mixed with the LAMP system and placed in the qPCR instrument for amplification (Fig. 4A). Sixteen of 132 urine samples were detected as positive by the extraction-free HF-LAMP assay (Fig. S3). The sensitivity of the extraction-free HF-LAMP assay compared with the reported qPCR assay was 55.5% (15/27). The reasons for relatively low sensitivity may include lower input amounts of samples (compared to those of the reactions with extracted DNA), low viral load in the samples, and the presence of some interfering substances for LAMP. When the Ct values of the samples were less than 25 and 30, the sensitivity of the extraction-free HF-LAMP assay was increased to 100.0% (6/6) and 75.0% (12/16, 95% confidence interval, 0.512−0.988) by SPSS 17.0 software, respectively (Fig. 4B), showing a significantly improved sensitivity. The specificity of the extraction-free HF-LAMP assay was 99.1% (106/107) (Fig. S3).

## Clinical evaluation of the nucleic acid extraction-free BKV HF-LAMP assay using clinical samples from kidney transplant patients

To further verify the clinical applicability of the nucleic acid extraction-free BKV HF-LAMP assay, 30 archived clinical urine samples from kidney transplant recipients were tested by

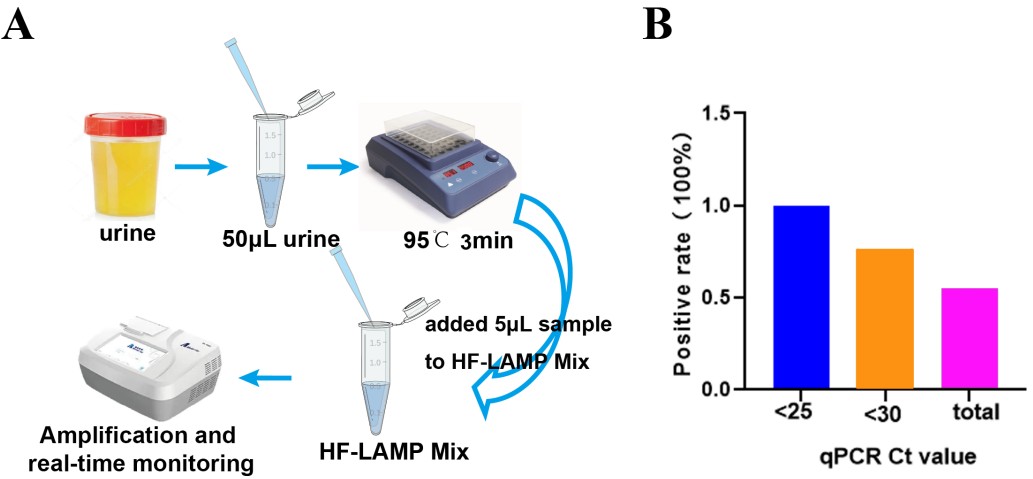

**Figure 4** **Clinical validation of the DNA extraction-free HF-LAMP method directly using urine samples.** (A) Workflow of DNA extraction-free HF-LAMP using urine samples. (B) Positive rate of the extraction-free HF-LAMP assay for all samples and those with Ct values less than 25 and 30 by the qPCR assay 2 (the previously reported qPCR assay) with extracted DNA.

the extraction-free HF-LAMP assay, the qPCR assay 1 (an approved commercial qPCR kit, (Sinomd Bio, Beijing, China)) and the qPCR assay 2 a previously reported qPCR assay (*Leung et al., 2001*), in parallel. HF-LAMP, the qPCR assay 1 and qPCR assay 2 detected 19, 19, and 17 out of 20 positive samples previously identified by a digital PCR assay, respectively (Table 2 and Fig. S4). The sensitivity, specificity and accuracy of the extraction-free HF-LAMP assay were 95%, 100%, and 96.7%, respectively, same to those of commercial qPCR kit but slightly higher than previously reported method (Table 3).

## POCT detection using a portable finger-driven microfluidic chip

To develop the POCT diagnosis, we further combined the HF-LAMP assay with a portable finger-driven microfluidic chip (Fig. 5) (*Wang et al., 2022*). An operator-blinded clinical trial was conducted to test 37 HIV-1 infected clinical urine samples, including 27 positive and 10 negative samples, which were confirmed by the previous qPCR assay (qPCR assay 2). The assay achieved a specificity of 80.0% (8/10) and a sensitivity of 74.1% (20/27) based on the 37 clinical samples (Table 4 and Table S2).

## DISCUSSION

BKV infection is ubiquitous, and the adult seropositivity rate is estimated to be 80–90% (*Antonsson et al., 2010*; *Egli et al., 2009*; *Kamminga et al., 2018*). BKV infection can cause significant morbidity in immunosuppressed individuals (*Reploeg, Storch & Clifford, 2001*). The incidences of BKV viremia and BKVN are 13% and 8% in the kidney transplant recipient population, respectively. In the absence of diagnosis and treatment, BKVN can lead to 80% graft loss (*Shen et al., 2021*). The 2019 guideline by the American Society of Transplantation Infectious Diseases Community of Practice suggested that all renal transplant recipients should be screened for blood BKV DNA monthly for nine

**Table 3** Comparison of the extraction-free HF-LAMP assay with two qPCR assays for 30 clinical samples from kidney transplant patients.

| Samples | HF-LAMP assay | | qPCR assay 1 (commercial qPCR kit, Sinomd) | | qPCR assay 2 (previously reported method) | |
|---|---|---|---|---|---|---|
| | Pos. | Neg. | Pos. | Neg. | Pos. | Neg. |
| Pos. ($n = 20$). | 19 | 1 | 19 | 1 | 18 | 2 |
| Neg. ($n = 10$) | 0 | 10 | 0 | 10 | 0 | 10 |
| Total | 19 | 11 | 19 | 11 | 18 | 12 |
| Sensitivity (%) | 95 | | 95 | | 90 | |
| Specificity (%) | 100 | | 100 | | 100 | |
| Accuracy (%) | 96.7 | | 96.7 | | 93.3 | |

**Notes.**

Pos., positive; Neg., negative.

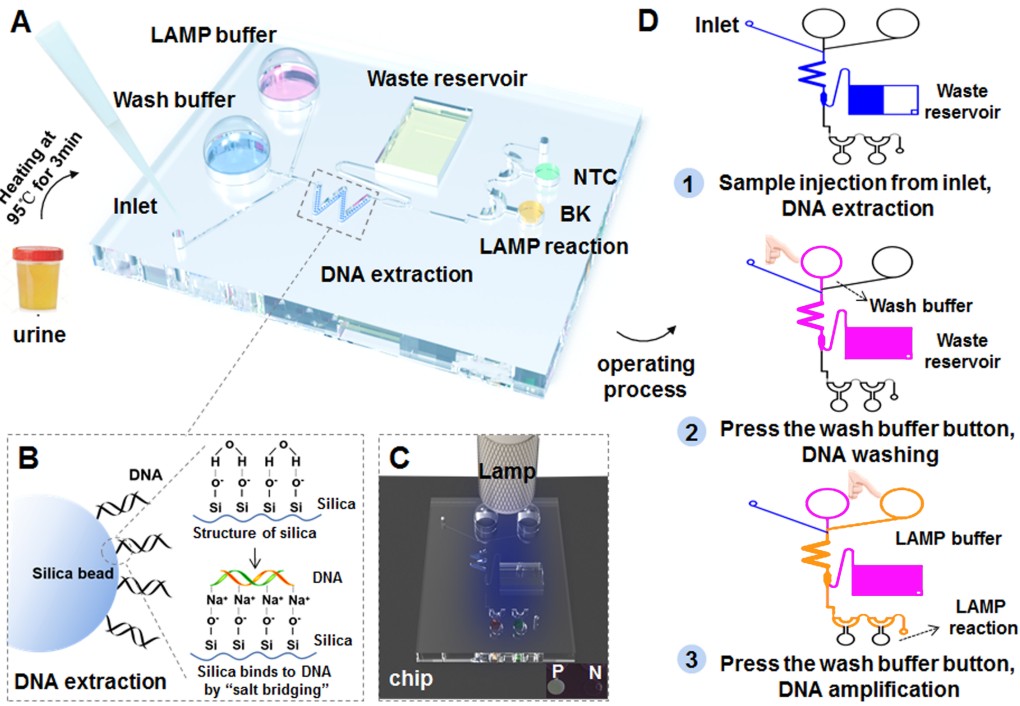

**Figure 5** **Workflow of the BKV HF-LAMP assay combined with the portable finger-driven microfluidic chip.** (A) The structure of the designed finger-driven microfluidic chip; (B) the working principle of the DNA extraction based on silica beads pre-filled into the chip; the wavy blue lines represent surface of silica beads. The figure on the left shows the phenomenon of DNA adsorption, while the figure on the right illustrates the whole principle. (C) The results of the HF-LAMP assay can be read by the fluorescence signal excited by blue lamp light; (D) the operation steps of the BKV detection based on the finger-driven microfluidic chip.

**Table 4  Comparison of the extraction-free HF-LAMP assay used the portable finger-driven microfluidic chip with qPCR assay 2 for 37 clinical samples from HIV-infected patients.**

| HF-LAMP assay | qPCR assay 2 (previously reported method) | | Total | Positive rate (%) | Concordance rate (%) |
|---|---|---|---|---|---|
| | Pos. | Neg. | | | |
| Pos. | 20 | 2 | 22 | 59.5 | 75.7 |
| Neg. | 7 | 8 | 15 | | |
| Total | 27 | 10 | 37 | | |
| Positive rate (%) | 73.0 | | | | |

**Notes.**

Pos., positive; Neg., negative.

consecutive months followed by every 3 months until 2 years post-transplant (*Kidney Disease: Improving Global Outcomes KDIGO Transplant Work Group, 2009*). Therefore, the development of a self-test method for BKV infection is essential for the prevention and control of BKV-associated diseases and deaths.

Although LAMP was developed over 20 years ago (*Notomi et al., 2000*), nonspecific amplification, low detection accuracy, and lack of multiplexing capacity have limited its application. Some efforts were made to mitigate and/or solve these problems and several probe-based LAMP methods were developed. For example, the HFman probe (*Dong et al., 2022*), molecular beacon (*Xu et al., 2022*), assimilating probe (*Lim et al., 2020*), and other strategies (*Ball et al., 2016*; *Tanner, Zhang & Evans Jr, 2012*) were used in LAMP to achieve specific and multiplex detection. In particular, the HFman probe-based real-time LAMP method (called HF-LAMP) uses a small amount of additional high-fidelity DNA polymerase and an HFman probe in a standard LAMP reaction system not only to achieve specific and multiplex detection but also to improve tolerance to primer-template mismatches caused by various virus variants (*Dong et al., 2022*; *Xu et al., 2021*; *Zhang et al., 2022*; *Zhou et al., 2019*; *Li et al., 2021*). Therefore, we applied HF-LAMP to develop a POCT assay for BKV infection.

The HF-LAMP assay was first established using extracted DNA. The assay showed a high specificity and sensitivity, and its LOD was 12 copies per 25 μL reaction. Its performance was further assessed by 132 urine samples from HIV-1 infected individuals. Patient urine and/or blood samples are usually used to detect BKV persistence in kidneys (*Kudose & Dong, 2014*). The direct use of urine as a sample for POCT will enable rapid and noninvasive detection of BKV (*Pinto et al., 2013*). Extracted nucleic acids from clinical samples are usually used in conventional testing workflows, which limits the implementation of POCT and leads to a delay in results. Notably, Bst DNA polymerase has good tolerance to urine components (*Jevtusevskaja et al., 2017*), and thermal lysis is a nucleic acid sample preparation technique without any other processing steps that has been used for urine samples analysis by LAMP (*Gandasegui et al., 2015*; *Packard et al., 2013*). Therefore, it is feasible for the BKV HF-LAMP assay to be performed with simple heating devices (*e.g.*, dry incubators or water baths), and urine can be directly used to detect the presence of BKV.

Therefore, we further developed the HF-LAMP assay to an extraction-free format, and evaluated its performance directly using clinical urine samples from HIV-1 infected individuals and kidney transplant recipients. BKVN is often associated with a higher BKV load ($>10^7$ copies/mL) in urine (*Hussain et al., 2020*). The detection sensitivity (100% for samples with BKV load of $>\sim 10^6$ copies/mL) of the extraction-free HF-LAMP assay enables the early identification in HIV-1 infected individuals with a potential risk of BKVN. For 30 archived clinical urine samples from kidney transplant recipients, the sensitivity, specificity and accuracy of the extraction-free HF-LAMP assay were 95%, 100%, and 96.7%, respectively. It turned out that extraction-free BKV HF-LAMP assay is a promising tool for monitoring BKV infection directly using urine samples. Furthermore, we combined the HF-LAMP assay with a portable finger-driven microfluidic chip to enable a more easy POCT diagnosis of BKV infection (*Wang et al., 2022*). Because of a relatively poor performance in sensitivity and specificity, possibly due to a low DNA capture rate of the chip and a weak fluorescence signal of FAM under a hand-held blue lamp, the current version of the portable finger-driven microfluidic chip might not suitable to be a diagnostic method. As a new attempt to POCT detection of BKV, however, the portable finger-driven microfluidic chip can be further improved and optimized in future.

## CONCLUSIONS

We developed an HF-LAMP assay for detection of BKV in urine, which shows a comparable specificity and sensitivity to two qPCR assays. Furthermore, an extraction-free format and a portable finger-driven microfluidic chip system of the HF-LAMP assay were developed for easy POCT diagnosis of BKV infection in urine.

### Funding

The study was supported by the Shanghai Science & Technology Innovation Action Program (21Y11900500) and the National Natural Science Foundation of China (81873621). The funders had no role in study design, data collection and analysis, decision to publish, or preparation of the manuscript.

### Grant Disclosures

The following grant information was disclosed by the authors:
Shanghai Science & Technology Innovation Action Program: 21Y11900500.
The National Natural Science Foundation of China: 81873621.

### Competing Interests

The authors declare there are no competing interests.

### Author Contributions

- Yongjuan Zhao performed the experiments, analyzed the data, prepared figures and/or tables, authored or reviewed drafts of the article, and approved the final draft.

- Yi Zeng performed the experiments, prepared figures and/or tables, and approved the final draft.
- Renfei Lu analyzed the data, authored or reviewed drafts of the article, and approved the final draft.
- Zhiying Wang performed the experiments, analyzed the data, prepared figures and/or tables, and approved the final draft.
- Xiaoling Zhang performed the experiments, prepared figures and/or tables, and approved the final draft.
- Nannan Wu analyzed the data, authored or reviewed drafts of the article, and approved the final draft.
- Tongyu Zhu analyzed the data, authored or reviewed drafts of the article, and approved the final draft.
- Yang Wang designed the experiments, supervised the study, analyzed the data, authored or reviewed drafts of the article, and approved the final draft.
- Chiyu Zhang conceived and designed the experiments, supervised the study, analyzed the data, authored or reviewed drafts of the article, and approved the final draft.

## Human Ethics

The following information was supplied relating to ethical approvals (i.e., approving body and any reference numbers):

The Shanghai Public Health Clinical Center granted Ethical approval to carry out the study within its facilities (2018Y032) and registered and recorded at the Chinese Clinical Trial Registry (ChiCTR1800017947).

## Data Availability

The raw data of LAMP and qPCR are available in the Supplemental Files.

## Supplemental Information

Supplemental information for this article can be found online at http://dx.doi.org/10.7717/peerj.14943#supplemental-information.

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
