# Peer review of "Rapid point-of-care detection of BK virus in urine by an HFman probe-based loop-mediated isothermal amplification assay and a finger-driven microfluidic chip"

_PeerJ, doi:10.7717/peerj.14943_

## Round 0.1 · original submission · Major Revisions

Please revise and resubmit at the earliest.

Reviewer 1 ·

Basic reporting

The manuscript is readable and clear with only occasional errors: Line 318 "directly sing", Line 73 "asymptomatic..." is a repeated sentence.

For references to literature, the authors do not sufficiently present the background of LAMP mutliplexing, claiming that their method was the first to enable this when that is not true (esp. Line 290), see for example: https://pubmed.ncbi.nlm.nih.gov/26980448/ and https://pubmed.ncbi.nlm.nih.gov/23030060/. Claiming LAMP wasn't used until COVID is not accurate, there were FDA-cleared assays from Meridian, for example. For the method of LAMP they cite one of the original LAMP papers and then their own paper from 2019 which is inappropriate, seems like a way to just get a citation for themselves and should instead use another original LAMP citation e.g. the Loop primer paper.

Tables are very hard to read, and should have lines separating the cells and data.

POCT is used without definition, and Ct/Tt are used without units which is a little confusing.

Experimental design

Research questions are good, plenty of samples and real-world utility with comparison to qPCR. The high-fidelity DNA polymerase is not defined, that should be fixed.

And the authors claim a waiver of informed consent, with documents attached to that effect. But the forms do not appear to be approved or signed.

Validity of the findings

Findings are mostly good and the assays work well. The biggest objection I have is that the authors claim a higher sensitivity than qPCR, but that's a bit of a stretch. Perhaps more samples read positive in the HF-LAMP, but when the direct samples are used the authors' data misses a significant number of the qPCR positive samples, with Figure 4 showing a detection of only 60%. The text should be more honest about this. Similarly, the authors state a detection of down to 3 copies but looking at Figure 2B only 1 of the replicates there is detected with reasonable signal and is far off the linearity of the higher inputs. No definition of LOD would allow for that and the authors should not claim it.

Reviewer 2 ·

Basic reporting

This manuscript by Zhao Y and colleagues describes a diagnostic method that has the potential to be used at the point of care (POC). They used the HFman probe-based LAMP and a figure-driven microfluidic chip. The manuscript is generally well written. The experiments were logically designed, and the data were reliable. However, some statements/conclusions are not fact-based. The manuscript must be improved before its publication.
First, this sentence in L#234-5 is wrong. “no samples were detected as negative by the HF-LAMP assay but positive by any one qPCR assays (Figure 3).” According to Fig 3, two such samples existed, they were #25 and #114. Both were tested negative by HF-LAMP, but positive by at least one PCR, i.e., both by qPCR assay 1.

Second, the sensitivity and specificity of the figure-driven microfluidic chip were only 74.1% and 80.0%, respectively, that is more than one quarter and one fifth sample would be false negative and false positive, which indicates that this method is not much valuable for being used as a diagnostic method. The authors thought that this poor performance was due to freeze-thaw of samples. It is the impression of this reviewer that all samples used in both HF-LAMP and figure-driven microfluidic chip were handled in the same way. Therefore, it is most likely the latter is not sensitive and specific in this case and is not suitable to be a diagnostic method.

Third, in line #282, the incidence is used. Is it incidence or prevalence? It seems too high to be incidence. Please check and make sure.
Fig 2 legend: how much DNA of other viruses were used? Please add here or in the main text.
Fig 4 legend: it should be B instead of C. In addition, sensitivity is used in the legend and positive rate (%) is used in the figure, all samples in the former and total in the latter. They should be consistent to avoid readers’ confusion.
L#237-8: the sentence belongs to discussion, not results.
L#231-3: rewrite the sentence. It does not flow well.
Minor points:
L#37: viability is not correct word.
L#40: Add “DNA” so it reads without DNA extraction.
L# 74: please define POCT since it is first use here.
L#115: information for MEGA software is needed.
L#124: it should be g/L, not g/mL.
L#128: it should be 10×, not 1×.
L#140: a period is needed so it reads: describe above.
L#201: did you mean identity instead of similarity?

Experimental design

See above

Validity of the findings

See above

·

Basic reporting

Other than a few typos, the English is pretty clear. BKV not my area but intro gave enough context to get on board so seems appropriate. Raw data available in supplemental file.

Experimental design

Research question defined (find better PoC assay for BKV). Methods could use a few additional details but are largely sufficient. Information on ethical review for clinical samples given.

Validity of the findings

Relatively large numbers of clinical samples tested. Data provided.

Additional comments

General comments:
This article reports the design of a HFman LAMP assay (addition of a TaqMan-like quencher-fluorophore probe and supplemental an error-correcting polymerase to standard LAMP) for BK virus. The authors design a primer set and probe and test their assay in clinical samples. They also provide preliminary data from implementing the assay in a microfluidic chip for potential point of care use. This appears to be the first report of sequence-specific LAMP detection for BKV and thus provides a useful resource both for researchers/clinicians implementing BKV testing and as an additional proof of principle for HF-LAMP.

One strong point of the paper is the use of a large number of clinical samples for validation of the method. However the methods, results and discussion could be polished somewhat to allow the reader to more easily follow these data.

One concern is that the two qPCR methods used here do not seem particularly concordant. Is this really the state of the art in the field? Is one qPCR more trustworthy than the other? Without having some ground truth to compare to, it becomes hard to interpret the results on the n=132 HIV-positive urine samples and the POCT testing. I think this needs to be dealt with more carefully in the text (lines 231-240 are a good start but should probably discuss a bit in discussion) and reporting sensitivity/specificity in this cohort with no gold standard positive/negative measurement seems problematic. If I understand things correctly, only the kidney cohort has an externally defined positive/negative diagnosis? If so, emphasis on this cohort should be enhanced since it's the only place with a (presumably) solid footing although sensitivity/specificity in the general kidney transplant population are then unclear since the cohort was specially selected.

Discussion could use some text on why HF-LAMP performed relatively poorly on extraction-free HIV samples but well on extraction-free kidney. Also did relatively poor performance in POCT just reflect extraction-free performance or was the POCT device degrading performance further? More direct comparison of extraction-free vs extracted DNA would be helpful to reader. Abstract should also mention the poorer performance in HIV cohort.

In addition, the relationship between the 30 kidney and 132 HIV samples used in initial testing and the n=37 urine samples used in POCT testing is unclear. This needs to be clarified before reader can interpret performance. And if the samples were tested in previous sections, then the performance should be compared to POCT performance. For example, if these samples were used in previous sections then there is presumably a LAMP Tt for both extracted DNA and for extraction-free urine and comparison to POCT +/- would be informative. Also if they were tested multiple times by qPCR (or had an external clinical call from kidney cohort) then that would help clarify the ambiguity in qPCR results. As is, the qPCR2 assay results are a somewhat problematic "gold standard" since they disagreed with qPCR1 in previous testing.

Also, no limit of detection was determined e.g. showing consistent detection at X copies/reaction. Without that, claims of detection down to X copies should be avoided.

A previous LAMP assay for BK virus has been reported:
https://doi.org/10.1128/jcm.01024-06
Seems worth citing and describing how this assay will improve upon.

Specific comments:
- line 42: "sensitivity (3 copies/reaction)" not standard definition of sensitivity and only one out of two reactions positive at 3 copies. Reword or additional experiments to clarify LOD.
- line 43: "showed a higher detection sensitivity" seems a bit ambiguous with uncertain samples. Perhaps "shower a equivalent and potentially even higher" or similar. Might also remove the numbers since "true positives" a bit unclear?
- line 47: probably should add a warning to reader "although some decreased performance was seen when performing extraction-free HF-LAMP on a cohort from individuals infected with HIV" or similar
- line 71: "lacks effective treatments" should probably should be "lacks effective antiviral treatments" since reduction of immunosuppression is treatment and also if there's no treatment then why bother to detect?
- line 73: BKV isn't my area but "when specific antiviral therapy and treatment strategy are not available" implies there are sometimes antiviral therapies available? Is that true? If so, it runs counter to the previous sentence, doesn't it? Perhaps "when" should be "until the development of" or else polish this section please.
- line 80: "qPCR ... requires specific PCR laboratories" doesn't really require a special "specific PCR" laboratory (e.g. various portable PCR machines or a PCR machine in any standard lab) so maybe remove the laboratory part.
- line 128: "2.5ul of 1x isothermal amplification buffer" presumably it was "10x amplification buffer if 2.5ul in 25ul reaction"? Either "1x isothermal amplification buffer" or "2.5ul of 10x isothermal amplification buffer". Also should probably define "isothermal amplification buffer" eithere company source or contents.
- line 129: "8mM MgSO4" isothermal amplification buffer usually has some Mg in it. Is this in addition to or including that?
- line 129: LF and LB concentrations listed but main primer set described here appears not to have LF. Add a "if used" caveat?
- line 164-179: Clearer to reader, if the cohorts are introduced in the same order as used in the results (currently HIV then Kidney)
- line 167: "Clinical urine samples were ... and do not need to be additionally provided. IRB waived the need for consent" I think I see what the authors are aiming for but could use rephrasing for clarity.
- line 172: "BKVN patients were diagnosed by immunohistochemistry" could use a bit more detail?
- line 181: What urine is being used here? Previously tested kidney/HIV or some new cohort? And what is the number tested?
- line 182: "those of a reported qPCR assay (Leung et al. 2001)" This sounds like the introduction of a new qPCR assay but this was already described on line 151. Maybe just "those from the qPCR assay" since already described above.
- line 186: Make clear whether a heating lysis step was used prior to adding urine to chip.
- line 187: "37 urine sample include 10 negative samples and 27 positive samples" Previously tested kidney/HIV or some new cohort? How was positive/negative determined?
- line 187: "with slight modifications" what are these modifications?
- line 197: "In this part, the experiment was conducted by Beihang University" It seems unusual to mention location of test. But since it's mentioned, I don't see any authors from Beihang. Maybe remove or clarify.
- line 196: "read the results directly by naked eyes under a hand-held blue lamp" Presumably there is also some sort of orange filtration to be able to see fluorescence (or lamp is UV instead of "blue"?)? Please clarify. Authors should also make clear that they're (presumably) using Fam labeled probes for eye detection and not Cy5.
- line 223: "The sensitivity of the LAMP assay was 3 copies/reaction in less than 20 min (Figure 2B)." Seems to mean something other than standard definition of "sensitivity" i.e. true positive rate. Perhaps "limit of detection"? But even then only one out of two 3 copy reactions was detected. Seems like need more reactions e.g. 20 and higher hit rate e.g. 95% if going to claim limit of detection down to 3 copies. Also not clear how many negative controls but would probably need additional negative controls at the same time e.g. 20. Please reword or add additional data/experiments?
- line 240: "the HF-LAMP assay showed higher concordance ... than commercial qPCR" reader can't really assess this from Table 1 since qPCR1 vs qPCR2 not compared. Add comparison to Table 1 or refer to Figure 3B.
- line 243: "high BKV" how is this defined?
- line 257: "25 (~10^6 copies/ml) and 30 (~10^5 copies/ml)" how were these Ct to copies/ml determined? Add to methods.
- line 266: "positive samples" perhaps "positive samples identified by previous digital PCR" if I understand correctly.
- line 277: "hand-held UV-lamp" wasn't this described as "blue lamp" on line 196? Clarify which.
- line 293: two out of three examples here are not SARS-CoV-2 making previous sentence a bit of a non-sequitur. Remove previous sentence or switch these to sequence-specific SARS-CoV-2 LAMP assays? There's plenty to cite e.g.
https://doi.org/10.1186/s13059-021-02387-y
https://doi.org/10.2144%2Fbtn-2020-0157
https://doi.org/10.3390/v13050742
- line 314: "of ~10^6 copies/ml" probably "of >~10^6 copies/ml"
- line 325: "slightly higher sensitivity" presumably this is based on HIV cohort since both LAMP and qPCR performed similar in kidney? If so, the "true" answer and significance of differences seems a bit nebulous so perhaps just "comparable specificity and sensitivity" would be more appropriate
- line 327: "for easy POCT diagnosis of BKV infection in urine at home" seems like "easy POCT diagnosis ... at home" is pushing it. Are these really ready to hand out to unskilled people e.g. how will reactions be incubated, how will thermal lysis be performed, how is appropriate volume of urine loaded, how are assays stored prior to use? Remove "at home" and seems more reasonable.
- line 335: data availability section formatted correctly?

- Table 1 caption: Maybe make clear these are all on "extracted DNA" so difference with Table 3 is clearer.
- Table 2: Are the 3 NAs adding anything? Just leave blank?
- Table 2: Is this redundant with Figure 3B? If not then seems like the two qPCR assays should also be compared? Could consider removing positive rate and/or totals if short on space (or clarify what value they are adding?).
- Table 3 caption: I think this is in POCT device? Make this clear.
- Table 3: The five NAs aren't really adding anything are they? Just leave blank?
- Figure 1: there's a little triangle just to the right of "5000". I guess that's probably "position 1" but why a triangle to indicate? Clarify in caption or figure presentation please.
- Figure 1A: What are "NCCR", "Agno", "tAg", "TAg", "VP1", "VP2","VP3" arrows? What are boxes with "0pqrs"? What does red/blue indicate for arrows? Clarify in caption please.
- Figure 2A: What is purple horizontal line? Presumably threshold. Describe in caption.
- Figure 2B: I would indicate the number of samples called negative by all three methods. Just a text "Negative by all three assays: XX" or a large circle around the whole thing with XX?
- Figure 3A: what is final row "other"? Probably supposed to show all samples labeled as negative by all three assays? Maybe label more specifically and also provide "(n=XX)".
- Figure 3A: I'm always a bit suspicious of plotting Ct and Tt on same scale. One cycle can <1 minute with fast ramp speed so the plot overstates the speed difference between the two assay types. But doesn't strongly affect results so if authors considered and want to present as such then fine.
- Figure 4 caption: "by the qPCR assay with extracted DNA" There's two qPCR assays reported previously, how were the Ct between the two combined to a single value?
- Figure 4A: Worth indicating that 5ul are added to HF-LAMP mix since specified 50ul earlier in figure?
- Figure 4B: Should also show the positivity rate in samples Negative by qPCR.
- Figure 4B: Could consider adding binomial proportion confidence interval e.g. https://en.wikipedia.org/wiki/Binomial_proportion_confidence_interval
- Figure 5 caption: "UV light" but methods list "blue lamp". Clarify.
- Figure 5B: Unclear what top right vs bottom right indicate. Unclear what wavy blue lines indicates. Mildly unclear how left and right side of figure relate.
- Figure 5C: "P N" down in bottom right corner pretty easy to miss. Make bigger and/or add "zoom in" lines to guide reader?
- Figure 5A: Is urine applied directly from collection tube? Or is there a heating step?
- Figure 5A/D: Why are there two amplification chambers? Are two different reactions being run? Were duplicate reactions run? If so, how was positivity assessed?
- Figure 5D: Blue/red/green coloring unclear. Indicate in caption what this is supposed to mean.
- Table S2: "+Pos/-Neg" column indicates some subjective call? What was the criteria? e.g. Samples 121 and 122 look pretty similar but have differing + and - call.
- Table S2: Add Tt from extraction-free and extracted DNA HF-LAMP so reader can compare
- Table S2: What qPCR value is being shown? If qPCR1, available could also show it.
- Figure S3: Might as well show original HF-LAMP on extracted DNA and qPCR1 too.
- Figure S3/S4 caption: Add "in HIV cohort"/"in Kidney cohort' (or similar) to make difference clear to reader

Typos/suggestions:
- line 56: "kb (kilobases)" "kb" never used again just write "kilobases" or "5000 bases"?
- line 86-87: "Hence, a rapid and convenient POCT assay ... without specifc PCR laboratories" Second sentence seems a bit redundant. Delete/combine?
- line 166: "we apply for" to "we applied for" also comma before we
- line 187: "include" to "including"
- line 188: "detected" to "tested" ("detected" mostly means positive result call to me but this sentence is describing -/+)
- line 189: "detection" to "manufacturing" or similar
- line 179/184: missing multiplication symbol "x" also "numbers" to "number" (twice)
- line: 195: "read the results" to "the results were read"
- line 196: "by naked eyes" to "by naked eye" or "by eye"
- line 228/229: "known as" to "here labeled" (twice)
- line 245: "indicat" to "indicate"
- line 263: "detected" to "tested"/"assayed"
- line 315: "identification of HIV-1 infected individuals" to "identification in HIV-1 infected individuals"
- line 318: "directly sing" to "directly using"
- Figure 1: "BK Genomen" presumably "BK Genome"
- Figure 3A: "No. of samples" to "No. of sample". Otherwise sounds like how many samples rather than sample ID. Also maybe just "ID of sample" to avoid ambiguity.
- Table 1 caption: Caption is repeated twice.
- Raw data tab "Figure 2B" cell A11: "0.00E+00" presumably "1.00E+00"?
- Figure S3: "Extracted RNA" presumably "Extracted DNA"

---

## Round 0.2 · Minor Revisions

Please revise as per the suggestions.

Reviewer 1 ·

Basic reporting

The authors have done a good job addressing the reviewers' comments from the first review, and have improved the manuscript significantly. My biggest concerns were how the assay performance was described, which has been improved, and the presentation of LAMP and the technology, which is also better. I have only a couple minor points but think the manuscript can be accepted:
1) The "LOD of 12 copies/reaction" is entirely an interpolation or estimation from the data. The authors show an LOD of 24 copies/reaction, potentially lower, but they shouldn't claim it unless proving it.
2) The error bars on Figure 4B are not described in the legend for the figure, and while partially described in the text a reader should be able to look at that figure and understand what it means. Also the Total bar seems to have a value of 0.6 +/- 0.6 or so which doesn't seem a particularly informative measure.

Experimental design

See above

Validity of the findings

See above

---

## Round 0.3 · accepted · Accept

Thank you for your response.